# MULTI-AGENT COLLABORATION VIA REWARD ATTRIBUTION DECOMPOSITION

## ABSTRACT

Recent advances in multi-agent reinforcement learning (MARL) have achieved super-human performance in games like Quake 3 and Dota 2. Unfortunately, these techniques require orders-of-magnitude more training rounds than humans and may not generalize to slightly altered environments or new agent configurations (i.e., *ad hoc team play*). In this work, we propose *Collaborative Q-learning* (**CollaQ**) that achieves state-of-the-art performance in the StarCraft multi-agent challenge and supports ad hoc team play. We first formulate multi-agent collaboration as a joint optimization on reward assignment and show that under certain conditions, each agent has a *decentralized Q*-function that is approximately optimal and can be decomposed into two terms: the *self-term* that only relies on the agent's own state, and the *interactive term* that is related to states of nearby agents, often observed by the current agent. The two terms are jointly trained using regular DQN, regulated with a *Multi-Agent Reward Attribution* (MARA) loss that ensures both terms retain their semantics. CollaQ is evaluated on various StarCraft maps, outperforming existing state-of-the-art techniques (i.e., QMIX, QTRAN, and VDN) by improving the win rate by $40\%$ with the same number of environment steps. In the more challenging ad hoc team play setting (i.e., reweight/add/remove units without re-training or finetuning), CollaQ outperforms previous SoTA by over $30\%$.

## 1 INTRODUCTION

In recent years, multi-agent deep reinforcement learning (MARL) has drawn increasing interest from the research community. MARL algorithms have shown super-human level performance in various games like Dota 2 (Berner et al., 2019), Quake 3 Arena (Jaderberg et al., 2019), and StarCraft (Samvelyan et al., 2019). However, the algorithms (Schulman et al., 2017; Mnih et al., 2013) are far less sample efficient than humans. For example, in Hide and Seek (Baker et al., 2019), it takes agents $2.69 - 8.62$ million episodes to learn a simple strategy of door blocking, while it only takes human several rounds to learn this behavior. One of the key reasons for the slow learning is that the number of joint states grows exponentially with the number of agents.

Moreover, many real-world situations require agents to adapt to new configurations of teams. This can be modeled as *ad hoc* multi-agent reinforcement learning (Stone et al., 2010) (Ad-hoc MARL) settings, in which agents must adapt to different team sizes and configurations at test time. In contrast to the MARL setting where agents can learn a fixed and team-dependent policy, in the Ad-hoc MARL setting agents must assess and adapt to the capabilities of others to behave optimally. Existing work in ad hoc team play either require sophisticated online learning at test time (Barrett et al., 2011) or prior knowledge about teammate behaviors (Barrett and Stone, 2015). As a result, they do not generalize to complex real-world scenarios. Most existing works either focus on improving generalization towards different opponent strategies (Lanctot et al., 2017; Hu et al., 2020) or simple ad-hoc setting like varying number of test-time teammates (Schwab et al., 2018; Long et al., 2020). We consider a more general setting where test-time teammates may have different capabilities. The need to reason about different team configurations in the Ad-hoc MARL results in an additional exponential increase (Stone et al., 2010) in representational complexity comparing to the MARL setting.

In the situation of collaboration, one way to address the complexity of the ad hoc team play setting is to explicitly model and address how agents collaborate. In this paper, one *key* observation is that when collaborating with different agents, an agent changes their behavior *because* she realizes that the team could function better if she focuses on some of the rewards while leaving other rewards to other teammates. Inspired by this principle, we formulate multi-agent collaboration as a joint

optimization over an implicit reward assignment among agents. Because the rewards are assigned differently for different team configurations, the behavior of an agent changes and adaptation follows.

While solving this optimization directly requires centralization at test time, we make an interesting theoretical finding that each agent has a *decentralized* policy that is **(1)** approximately optimal for the joint optimization, and **(2)** only depends on the local configuration of other agents. This enables us to learn a direct mapping from states of nearby agents (or "observation" of agent $i$) to its $Q$-function using deep neural network. Furthermore, this finding also suggests that the $Q$-function of agent $i$ should be decomposed into two terms: $Q_i^{\text{alone}}$ that only depends on agent $i$'s own state $s_i$, and $Q_i^{\text{collab}}$ that depends on nearby agents but vanishes if no other agents nearby. To enforce this semantics, we regularize $Q_i^{\text{collab}}(s_i, \cdot) = 0$ in training via a novel ***Multi-Agent Reward Attribution*** (MARA) loss.

The resulting algorithm, ***Collaborative Q**-learning* (CollaQ), achieves a 40% improvement in win rates over state-of-the-art techniques for the StarCraft multi-agent challenge. We show that **(1)** the MARA Loss is critical for strong performance and **(2)** both $Q^{\text{alone}}$ and $Q^{\text{collab}}$ are interpretable via visualization. Furthermore, CollaQ agents can achieve *ad hoc team play* without retraining or fine-tuning. We propose three tasks to evaluate ad hoc team play performance: at test time, **(a)** assign a new VIP unit whose survival matters, **(b)** swap different units in and out, and **(c)** add or remove units. Results show that CollaQ outperforms baselines by an average of 30% in all these settings.

**Related Works.** The most straightforward way to train such a MARL task is to learn individual agent's value function $Q_i$ independently(IQL) (Tan, 1993). However, the environment becomes non-stationary from the perspective of an individual agent thus this performs poorly in practice. Recent works, e.g., VDN (Sunehag et al., 2017), QMIX (Rashid et al., 2018), QTRAN (Son et al., 2019), adopt centralized training with decentralized execution to solve this problem. They propose to write the joint value function as $Q^\pi(s, \mathbf{a}) = \phi(s, Q_1(o_1, a_1), ..., Q_K(o_K, a_K))$ but the formulation of $\phi$ differs in each method. These methods successfully utilize the centralized training technique to alleviate the non-stationary issue. However, none of the above methods generalize well to ad-hoc team play since learned $Q_i$ functions highly depend on the existence of other agents.

## 2 COLLABORATIVE MULTI-AGENT REWARD ASSIGNMENT

**Basic Setting**. A multi-agent extension of Markov Decision Process called collaborative partially observable Markov Games (Littman, 1994), is defined by a set of states $S$ describing the possible configurations of all $K$ agents, a set of possible actions $A_1, \ldots, A_K$, and a set of possible observations $O_1, \ldots, O_K$. At every step, each agent $i$ chooses its action $a_i$ by a stochastic policy $\pi_i : O_i \times A_i \to [0, 1]$. The joint action $\mathbf{a}$ produces the next state by a transition function $P : S \times A_1 \times \cdots \times A_K \to S$. All agents share the same reward $r : S \times A_1 \times \cdots \times A_K \to \mathbb{R}$ and with a joint value function $Q^\pi = \mathbb{E}_{s_{t+1:\infty}, \mathbf{a}_{t+1:\infty}}[R_t | s_t, \mathbf{a}_t]$ where $R_t = \sum_{j=0}^\infty \gamma^j r_{t+j}$ is the discounted return.

In Sec. 2.1, we first model multi-agent collaboration as a joint optimization on reward assignment: instead of acting based on the joint state $\mathbf{s}$, each agent $i$ is acting *independently* on its own state $s_i$, following its own optimal value $V_i$, which is a function of the *perceived reward assignment* $\mathbf{r}_i$. While the optimal perceived reward assignment $\mathbf{r}_i^*(\mathbf{s})$ depends on the joint state of all agents and requires centralization, in Sec. 2.2, we prove that there exists an approximate optimal solution $\hat{\mathbf{r}}_i$ that only depends on the local observation $\mathbf{s}_i^{\text{local}}$ of agent $i$, and thus enabling *decentralized* execution. Lastly in Sec. 2.3, we distill the theoretical insights into a practical algorithm CollaQ, by directly learning the compositional mapping $\mathbf{s}_i^{\text{local}} \mapsto \hat{\mathbf{r}}_i \mapsto V_i$ in an end-to-end fashion, while keeping the decomposition structure of self state and local observations.

### 2.1 BASIC ASSUMPTION

A naive modeling of multi-agent collaboration is to estimate a joint value function $V_{\text{joint}} := V_{\text{joint}}(s_1, s_2, \ldots, s_K)$, and find the best action for agent $i$ to maximize $V_{\text{joint}}$ according to the current joint state $\mathbf{s} = (s_1, s_2, \ldots, s_K)$. However, it has three fundamental drawbacks: **(1)** $V_{\text{joint}}$ generally requires exponential number of samples to learn; **(2)** in order to evaluate this function, a full observation of the states of *all* agents is required, which disallows decentralized execution, one key preference of multi-agent RL; and **(3)** for any environment/team changes (e.g., teaming with different agents), $V_{\text{joint}}$ needs to be relearned for all agents and renders ad hoc team play impossible.

Our CollaQ addresses the three issues with a novel theoretical framework that decouples the interactions between agents. Instead of using $V_{\text{joint}}$ that bundles all the agent interactions together, we consider the underlying *mechanism* how they interact: in a fully collaborative setting, the reason why

agent $i$ takes actions towards a state, is not only because that state is rewarding to agent $i$, but also because it is *more* rewarding to agent $i$ than other agents in the team, from agent $i$'s point of view. This is the concept of *perceived reward* of agent $i$. Then each agent acts independently following its own value function $V_i$, which is the optimal solution to the Bellman equation conditioned on the assigned perceived reward, and is a function of it. This naturally leads to collaboration.

We build a mathematical framework to model such behaviors. Specifically, we make the following assumption on the behavior of each agent:

**Assumption 1.** *Each agent $i$ has a **perceived** reward assignment $\mathbf{r}_i \in \mathbb{R}_+^{|S_i||A_i|}$ that may depend on the joint state $\mathbf{s} = (s_1, \ldots, s_K)$. Agent $i$ acts according to its own state $s_i$ and individual optimal value $V_i = V_i(s_i; \mathbf{r}_i)$ (and associated $Q_i(s_i, a_i; \mathbf{r}_i)$), which is a function of $\mathbf{r}_i$.*

Note that the perceived reward assignment $\mathbf{r}_i \in \mathbb{R}_+^{|S_i||A_i|}$ is a non-negative vector containing the assignment of scalar reward at each state-action pair (hence its length is $|S_i||A_i|$). We might also equivalently write it as a function: $r_i(x, a) : S_i \times A_i \mapsto \mathbb{R}$, where $x \in S_i$ and $a \in A_i$. Here $x$ is a dummy variable that runs through all states of agent $i$, while $s_i$ refers to its current state.

Given the perceived rewards assignment $\{\mathbf{r}_i\}$, the values and actions of agents become *decoupled*. Due to the fully collaborative nature, a natural choice of $\{\mathbf{r}_i\}$ is the optimal solution of the following objective $J(\mathbf{r}_1, \mathbf{r}_2, \ldots, \mathbf{r}_K)$. Here $\mathbf{r}_e$ is the external rewards of the environment, $\mathbf{w}_i \geq 0$ is the preference of agent $i$ and $\odot$ is the Hadamard (element-wise) product:

$$J(\mathbf{r}_1, \ldots, \mathbf{r}_K) := \sum_{i=1}^K V_i(s_i; \mathbf{r}_i) \qquad \text{s.t.} \quad \sum_{i=1}^K \mathbf{w}_i \odot \mathbf{r}_i \leq \mathbf{r}_e \tag{1}$$

Note that the constraint ensures that the objective has bounded solution. Without this constraints, we could easily take each perceived reward $\mathbf{r}_i$ to $+\infty$, since each value function $V_i(s_i; \mathbf{r}_i)$ monotonously increases with respect to $\mathbf{r}_i$. Intuitively, Eqn. 1 means that we "assign" the external rewards $\mathbf{r}_e$ optimally to $K$ agents as perceived rewards, so that their overall values are the highest.

In the case of sparse reward, most of the state-action pair $(x, a)$, $r_e(x, a) = 0$. By Eqn. 1, for all agent $i$, their perceived reward $r_i(x, a) = 0$. Then we only focus on nonzero entries for each $\mathbf{r}_i$. Define $M$ to be the number of state-action pairs with positive reward: $M = \sum_{a_i \in A_i} \mathbb{1}\{r_i(x, a_i) > 0\}$. Discarding zero-entries, we could regard all $\mathbf{r}_i$ as $M$-dimensional vector. Finally, we define the reward matrix $R = [\mathbf{r}_1, \ldots, \mathbf{r}_K] \in \mathbb{R}^{M \times K}$.

**Clarification on Rewards**. There are two kinds of rewards here: external reward $r_e$ and perceived reward for each agent $r_i$. $r_e$ is defined to be the environmental reward shared by all the agents: $r_e : S \times A_1 \times \cdots \times A_k \to R$. Given this external reward, depending on a specific reward assignment, each agent can receive a perceived reward $r_i$ that drives its behavior. If the reward assignment is properly defined/optimized, then all the agents can act based on the perceived reward to jointly optimize (maximize) the shared external reward.

## 2.2 Learn to Predict the Optimal Assigned Reward $\mathbf{r}_i^*(\mathbf{s})$

The optimal reward assignments $R^*$ of Eq. 1, as well as its $i$-th assignment $\mathbf{r}_i^*$, is a function of the joint states $\mathbf{s} = \{s_1, s_2, \ldots, s_K\}$. Once the optimization is done, each agent can get the best action $a_i^* = \arg\max_{a_i} Q_i(s_i, a_i; \mathbf{r}_i^*(\mathbf{s}))$ independently from the reconstructed $Q$ function.

The formulation $V_i(s_i; \mathbf{r}_i)$ avoids learning the value function of statistically infeasible joint states $V_i(\mathbf{s})$. Since an agent acts solely based on $\mathbf{r}_i$, ad hoc team play becomes possible if the correct $\mathbf{r}_i$ is assigned. However, there are still issues. First, since each $V_i$ is a convex function regarding $\mathbf{r}_i$, maximizing Eqn. 1 is a summation of convex functions under linear constraints optimization, and is hard computationally. Furthermore, to obtain actions for each agent, we need to solve Eqn. 1 at every step, which still requires centralization at test time, preventing us from decentralized execution.

To overcome optimization complexity and enable decentralized execution, we consider *learning* a direct mapping from the joint state $\mathbf{s}$ to optimally assigned reward $\mathbf{r}_i^*(\mathbf{s})$. However, since $\mathbf{s}$ is a joint state, learning such a mapping can be as hard as modeling $V_i(\mathbf{s})$.

Fortunately, $V_i(s_i; \mathbf{r}_i(\mathbf{s}))$ is not an arbitrary function, but the optimal value function that satisfies Bellman equation. Due to the speciality of $V_i$, we could find an approximate assignment $\hat{\mathbf{r}}_i$ for each agent $i$, so that $\hat{\mathbf{r}}_i$ only depends on a *local observation* $\mathbf{s}_i^{\text{local}}$ of the states of nearby other agents

observed by agent $i$: $\hat{\mathbf{r}}_i(\mathbf{s}) = \hat{\mathbf{r}}_i(\mathbf{s}_i^{\text{local}})$. At the same time, these approximate reward assignments $\{\hat{\mathbf{r}}_i\}$ achieve approximate optimal for the joint optimization (Eqn. 1) with bounded error:

**Theorem 1.** *For all $i \in \{1, \ldots, K\}$, all $s_i \in S_i$, there exists a reward assignment $\hat{\mathbf{r}}_i$ that (1) only depends on $\mathbf{s}_i^{\text{local}}$ and (2) $\hat{\mathbf{r}}_i$ is the $i$-th column of a feasible global reward assignment $\hat{R}$ such that*

$$J(\hat{R}) \geq J(R^*) - (\gamma^C + \gamma^D)R_{\max}MK, \tag{2}$$

*where $C$ and $D$ are constants related to distances between agents/rewards (details in Appendix).*

Since $\hat{\mathbf{r}}_i$ only depends on the local observation of agent $i$ (i.e., agent's own state $s_i$ as well as the states of nearby agents), it enables *decentralized execution*: for each agent $i$, the local observation is sufficient for an agent to act near optimally.

**Limitation**. One limitation of Theorem 1 is that the optimality gap of $\hat{\mathbf{r}}_i$ heavily depends on the size of $\mathbf{s}_i^{\text{local}}$. If the local observation of agent $i$ covers more agents, then the gap is smaller but the cost to learn such a mapping is higher, since the mapping has more input states and becomes higher-dimensional. In practice, we found that the observation $o_i$ of agent $i$ covers $\mathbf{s}_i^{\text{local}}$ works sufficiently well, as shown in the experiments (Sec. 4).

## 2.3 Collaborative Q-Learning (CollaQ)

While Theorem. 1 shows the *existence* of perceived reward $\hat{\mathbf{r}}_i = \hat{\mathbf{r}}_i(\mathbf{s}_i^{\text{local}})$ with good properties, learning $\hat{\mathbf{r}}_i(\mathbf{s}_i^{\text{local}})$ is not a trivial task. Learning it in a supervised manner requires (close to) optimal assignments as the labels, which in turn requires solving Eqn. 1. Instead, we resort to an end-to-end learning of $Q_i$ for each agent $i$ with proper decomposition structure inspired by the theory above.

To see this, we expand the $Q$-function for agent $i$: $Q_i = Q_i(s_i, a_i; \hat{\mathbf{r}}_i)$ with respect to its perceived reward. We use a Taylor expansion at the *ground-zero* reward $\mathbf{r}_{0i} = \mathbf{r}_i(s_i)$, which is the perceived reward when only agent $i$ is present in the environment:

$$Q_i(s_i, a_i; \hat{\mathbf{r}}_i) = \underbrace{Q_i(s_i, a_i; \mathbf{r}_{0i})}_{Q^{\text{alone}}(s_i, a_i)} + \underbrace{\nabla_{\mathbf{r}} Q_i(s_i, a_i; \mathbf{r}_{0i}) \cdot (\hat{\mathbf{r}}_i - \mathbf{r}_{0i}) + \mathcal{O}(\|\hat{\mathbf{r}}_i - \mathbf{r}_{0i}\|^2)}_{Q^{\text{collab}}(\mathbf{s}_i^{\text{local}}, a_i)} \tag{3}$$

Here $Q_i(s_i, a_i; \mathbf{r}_{0i})$ is the *alone* policy of an agent $i$. We name it $Q^{\text{alone}}$ since it operates as if other agents do not exist. The second term is called $Q^{\text{collab}}$, which models the interaction among agents via perceived reward $\hat{\mathbf{r}}_i$. Both $Q^{\text{alone}}$ and $Q^{\text{collab}}$ are neural networks. Thanks to Theorem 1, we only need to feed local observation $o_i := \mathbf{s}_i^{\text{local}}$ of agent $i$, which contains the observation of $W < K$ local agents (Fig. 1), for an approximate optimal $Q_i$. Then the overall $Q_i$ is computed by a simple addition (here $o_i^{\text{alone}} := s_i$ is the individual state of agent $i$):

$$Q_i(o_i, a_i) = Q_i^{\text{alone}}(o_i^{\text{alone}}, a_i) + Q_i^{\text{collab}}(o_i, a_i) \tag{4}$$

**Multi-Agent Reward Attribution (MARA) Loss**. With a simple addition, the solution of $Q_i^{\text{alone}}$ and $Q_i^{\text{collab}}$ might not be unique: indeed, we might add any constant to $Q^{\text{alone}}$ and subtract that constant from $Q^{\text{collab}}$ to yield the same overall $Q_i$. However, according to Eqn. 3, there is an additional constraint: if $o_i = o_i^{\text{alone}}$ then $\hat{\mathbf{r}}_i = \mathbf{r}_{0i}$ and $Q^{\text{collab}}(o_i^{\text{alone}}, a_i) \equiv 0$, which eliminates such an ambiguity. For this, we add Multi-agent Reward Attribution (MARA) Loss.

**Overall Training Paradigm**. For agent $i$, we use standard DQN training with MARA loss. Define $y = \mathbb{E}_{s' \sim \varepsilon}[r + \gamma \max_{a'} Q_i(o', a')|s, a]$ to be the target $Q$-value, the overall training objective is:

$$L = \mathbb{E}_{s_i, a_i \sim \rho(\cdot)}[\underbrace{(y - Q_i(o_i, a_i))^2}_{\text{DQN Objective}} + \underbrace{\alpha(Q_i^{\text{collab}}(o_i^{\text{alone}}, a_i))^2}_{\text{MARA Objective}}] \tag{5}$$

where the hyper-parameter $\alpha$ determines the relative importance of the MARA objective against the DQN objective. We observe that with MARA loss, training is much stabilized. We use a soft constraint version of MARA Loss. To train multiple agents together, we follow QMIX and feed the output of $\{Q_i\}$ into a top network and train in an end-to-end centralized fashion.

CollaQ has advantages compared to normal Q-learning. Since $Q_i^{\text{alone}}$ only takes $o_i^{\text{alone}}$ whose dimension is independent of the number of agents, this term can be learned exponentially faster than $Q_i^{\text{collab}}$. Thus, CollaQ enjoys a much faster learning speed as shown in Fig. 5, Fig. 6 and Fig. 7.

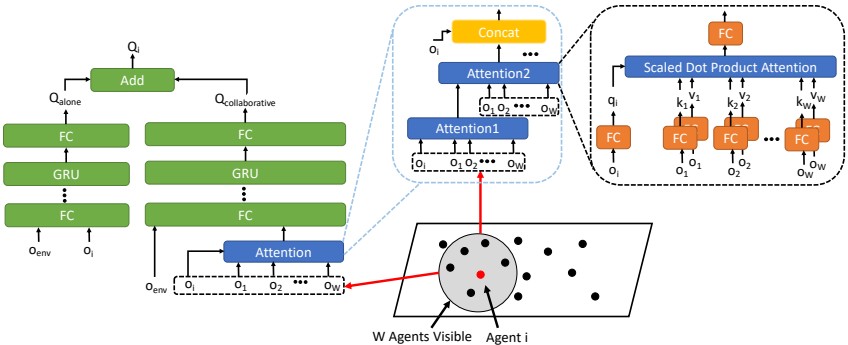

Figure 1: Architecture of the network. We use normal DRQN architecture for $o_i^{alone}$ with attention-based model for $Q^{\mathrm{collab}}$. The attention layers take the encoded inputs from all agents and output an attention embedding.

**Attention-based Architecture**. Fig. 1 illustrates the overall architecture. For agent $i$, the local observation $o_i := \mathbf{s}_i^{\mathrm{local}}$ is separated into two parts, $o_i^{\mathrm{alone}} := s_i$ and $o_i = \mathbf{s}_i^{\mathrm{local}}$. Here, $o_i^{\mathrm{alone}}$ is sent to the left tower to obtain $Q^{\mathrm{alone}}$, while $o_i$ is sent to the right tower to obtain $Q^{\mathrm{collab}}$. We use attention architecture between $o_i^{\mathrm{alone}}$ and other agents' states in the field of view of agent $i$. This is because the observation $o_i$ can be spatially large and cover agents whose states do not contribute much to agent $i$'s action, and effective $\mathbf{s}_i^{\mathrm{local}}$ is smaller than $o_i$. Our architecture is similar to EPC (Long et al., 2020) except that we use a transformer architecture (stacking multiple layers of attention modules). As shown in the experiments, this helps improve the performance in various StarCraft settings.

**Intuition of CollaQ and Connection to the Theory**. The intuitive explanation to CollaQ and MARA Loss is that when the agent cannot see others (i.e., other agent has no influence on the particular agent), the Q-value $Q_i$ should be equal to individual Q-value $Q_i^{\mathrm{alone}}$. This can be interpreted as some equivalent statements: 1. The problem can be decomposed well into local sub-problems. 2. The existence of other agents does not influence the Q-value of the particular agent. The inspired MARA loss helps to eliminate the ambiguity. The semantic meaning of $Q_i^{\mathrm{alone}}$ and $Q_i$ are shown in Fig. 3.

The intuition actually connects to the theory. Theorem 1. shows that under some mild assumptions, the CollaQ objective can be viewed as a sub-optimal solution to an optimization problem on reward assignment. Thus each component of CollaQ and MARA loss can be well-justified. Although the problem defined in Eq. 1 is hard to optimize, the empirical success of CollaQ to some extent shows the effectiveness. The theory here serves more as an inspiration to the practical algorithm. We leave the analysis between exact optimization and CollaQ to future work.

## 3 EXPERIMENTS ON RESOURCE COLLECTION

In this section, we demonstrate the effectiveness of CollaQ in a toy gridworld environment where the states are fully observable. We also visualize the trained policy $Q_i$ and $Q_i^{\mathrm{alone}}$.

**Ad hoc Resource Collection**. We demonstrate CollaQ in a toy example where multiple agents collaboratively collect resources from a grid world to maximize the aggregated team reward. In this setup, the same type of resources can return different rewards depending on the type of agent that collects it. The reward setup is randomly initialized at the beginning of each episode and can be seen by all the agents. The game ends when all the resources are collected. An agent is *expert* for a certain resource if it gets the highest reward among the team collecting that. As a consequence, to maximize the shared team reward, the optimal strategy is to let the *expert* collect the corresponding resource.

For testing, we devise the following reward setup: We have apple and lemon as our resources and $N$ agents. For picking lemon, agent 1 receives the highest reward for the team, agent 2 gets the second highest, and so on. For apple, the reward assignment is reversed (agent $N$ gets the highest reward, agent $N-1$ gets the second

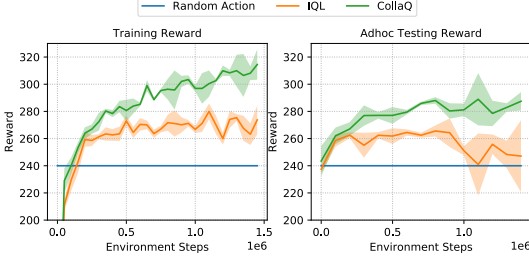

Figure 2: Results in resource collection. CollaQ (green) produces much higher rewards in both training and ad hoc team play than IQL (orange).

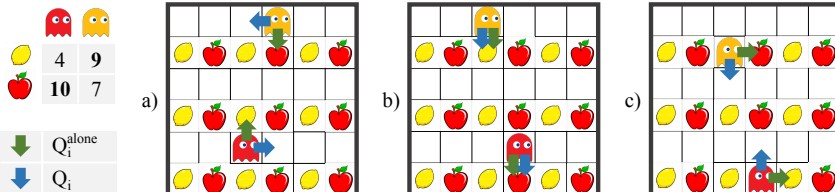

Figure 3: Visualization of $Q_i^{\text{alone}}$ and $Q_i$ in resource collection. The reward setup is shown in the leftmost column. Interesting behaviors emerge: in b), $Q_i^{\text{collab}}$ reinforces the behavior of $Q_i^{\text{alone}}$ since they are both the expert for the nearest resources; in a) and c), $Q_i^{\text{collab}}$ alters the decision of collecting lemon for red agent since it has lower reward for lemon compared with the yellow agent and similar phenomena occurs for the yellow agent.

highest, ...). This specific reward setup is excluded from the environment setup for training. This is a very hard ad hoc team play at test time since the agents need to demonstrate completely different behaviors from training time to achieve a higher team reward.

The left figure in Fig. 2 shows the training reward and the right one shows the ad hoc team play. We train on 5 agents in this setting. CollaQ outperforms IQL in both training and testing. In this example, random actions work reasonably well. Any improvement over it is substantial.

**Visualization of $Q_i^{\text{alone}}$ and $Q_i$.** In Fig. 3, we visualize the trained $Q_i^{\text{alone}}$ and $Q_i$ (the overall policy for agent $i$) to show how $Q_i^{\text{collab}}$ affects the behaviors of each agent. The policies $Q_i^{\text{alone}}$ and $Q_i$ learned by CollaQ are both meaningful: $Q_i^{\text{alone}}$ is the simple strategy of collecting the nearest resource (the optimal policy when the agent is the only one acting in the environment) and $Q_i$ is the optimal policy described formerly.

The leftmost column in Fig. 3 shows the reward setup for different agents on collecting different resources (e.g. the red agent gets 4 points collecting lemon and gets 10 points collecting apple). The red agent specializes at collecting apple and the yellow specializes at collecting lemon. In a), $Q_i^{\text{alone}}$ directs both agents to collect the nearest resource. However, neither agent is the expert on collecting its nearest resource. Therefore, $Q_i^{\text{collab}}$ alters the decision of $Q_i^{\text{alone}}$, directing $Q_i$ towards resources with the highest return. This behavior is also observed in c) with a different resource placement. b) shows the scenario where both agents are the expert on collecting the nearest resource. $Q_i^{\text{collab}}$ reinforces the decision of $Q_i^{\text{alone}}$, making $Q_i$ points to the same resource as $Q_i^{\text{alone}}$.

## 4 EXPERIMENTS ON STARCRAFT MULTI-AGENT CHALLENGE

StarCraft multi-agent challenge (Samvelyan et al., 2019) is a widely-used benchmark for MARL evaluation. The task in this environment is to manage a team of units (each unit is controlled by an agent) to defeat the team controlled by build-in AIs. While this task has been extensively studied in previous works, the performance of the agents trained by the SoTA methods (e.g., QMIX) deteriorates with a slight modification to the environment setup where the *agent IDs* are changed. The SoTA methods severely overfit to the precise environment and thus cannot generalize well to ad hoc team play. In contrast, CollaQ has shown better performance in the presence of random agent IDs, generalizes significantly better in more diverse test environments (e.g., adding/swapping/removing a unit at test time), and is more robust in ad hoc team play.

### 4.1 ISSUES IN THE CURRENT BENCHMARK

In the default StarCraft multi-agent environment, the ID of each agent never changes. Thus, a trained agent can memorize what to do based on its ID instead of figuring out the role of its units dynamically during the play. As illustrated in Fig. 4, if we randomly shuffle the IDs of the agents at test time, the performance of QMIX gets much worse. In some cases (e.g., 8m_vs_9m), the win rate drops from 95% to 50%, deteriorating by more than 40%. The results show that QMIX relies on the extra information (the order of agents) for generalization. As a consequence, the resulting agents overfit to the exact setting, making it less robust in ad hoc team play. Introducing random shuffled agent IDs at training time addresses this issue for QMIX as illustrated in Fig. 4.

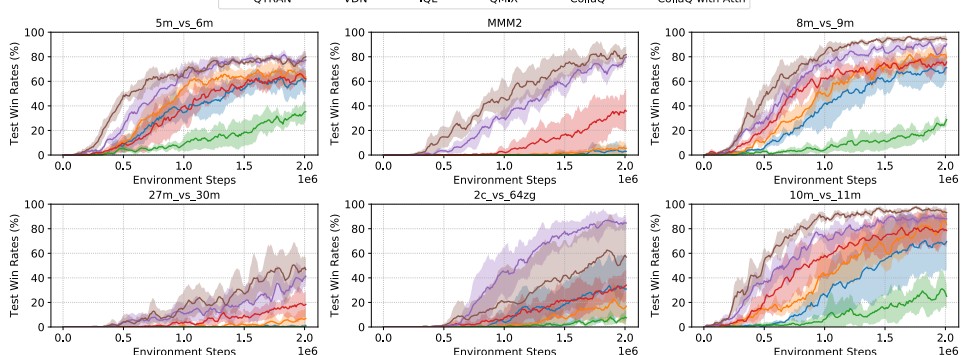

Figure 4: QMIX overfits to agent IDs. Introducing random agent IDs at test time greatly affect the performance.

Figure 5: Results in standard StarCraft benchmarks with random agent IDs. CollaQ (without Attn and with Attn) clearly surpasses the previous SoTAs. The attention-based model further improves the win rates for all maps except 2c_vs_64zg, which only has 2 agents and attention may not bring up enough benefits.

## 4.2  STARCRAFT MULTI-AGENT CHALLENGE WITH RANDOM AGENT IDS

Since using random IDs facilitates the learning of different roles, we perform extensive empirical study under this setting. We show that CollaQ on multiple maps in StarCraft outperforms existing approaches. We use the hard scenarios (e.g., 27m_vs_30m, MMM2 and 2c_vs_64zg) since they are largely unsolved by previous methods. Maps like 10m_vs_11m, 5m_vs_6m and 8m_vs_9m are considered medium difficult. For completeness, we also provide performance comparison under the regular setting in Appendix D Fig. 10. As shown in Fig. 5, CollaQ outperforms multiple baselines (QMIX, QTRAN, VDN, and IQL) by around 30% in terms of win rate in multiple hard scenarios. With attention model, the performance is even stronger.

Trained CollaQ agents demonstrate interesting behaviors. On MMM2: (1) Medivac dropship only heals the unit under attack, (2) damaged units move backward to avoid focused fire from the opponent, while healthy units move forward to undertake fire. In comparison, QMIX only learns (1) and it is not obvious (2) was learned. On 2c_vs_64zg, CollaQ learns to focus fire on one side of the attack to clear one of the corridors. It also demonstrates the behavior to retreat along that corridor while attacking while agents trained by QMIX does not. See Appendix D for more video snapshots.

## 4.3  AD HOC TEAM WORK

Now we demonstrate that CollaQ is robust to change of agent configurations and/or priority during test time, i.e., ad hoc team play, in addition to handling random IDs.

**Different VIP agent**. In this setting, the team would get an additional reward if the VIP agent is alive after winning the battle. The VIP agent is randomly selected from agent 1 to $N-1$ during training. At test time, agent $N$ becomes the VIP, which is a new setup that is not seen in training. Fig. 6 shows the VIP agent survival rate at test time. We can see that CollaQ outperforms QMIX by 10%-32%. We also see that CollaQ learns the behavior of *protecting VIP*: when the team is about to win, the VIP agent is covered by other agents to avoid being attacked. Such behavior is not clearly shown in QMIX when the same objective is presented.

**Swap / Add / Remove different units**. We also test the ad hoc team play in three harder settings: we swap the agent type, add and remove one agent at test time. From Fig. 7, we can see that CollaQ can generalize better to the ad hoc test setting. Note that to deal with the changing number of agents at test time, all of the methods (QMIX, QTRAN, VDN, IQL, and CollaQ) are augmented with

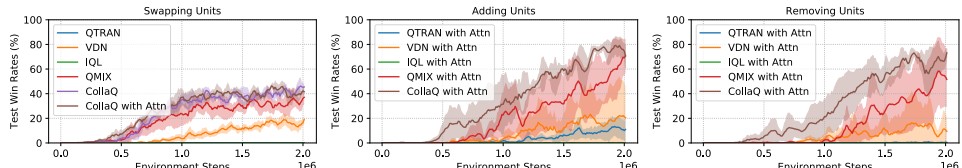

Figure 6: Results for StarCraft ad hoc team play using different VIP agent. At test time, the CollaQ has substantially higher VIP survival rate than QMIX. Attention-based model also boosts up the survival rate.

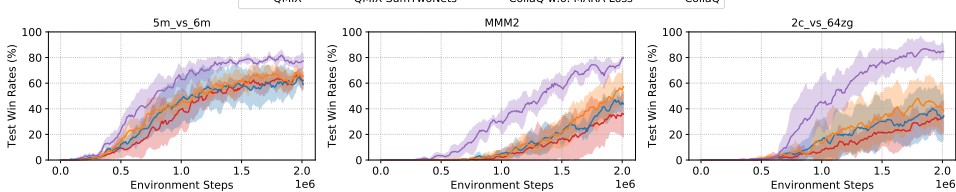

Figure 7: Ad hoc team play on: a) swapping, b) adding, and c) removing a unit at test time. CollaQ outperforms QMIX and other methods substantially on all these 3 settings.

Figure 8: Ablation studies on the mixture of experts and the effect of MARA Loss. CollaQ outperforms QMIX with a mixture of experts by a large margin and removing MARA Loss significantly degrades the performance.

attention-based neural architectures for a fair comparison. We can also see that CollaQ outperforms QMIX, the second best, by $9.21\%$ on swapping, $14.69\%$ on removing, and $8.28\%$ on adding agents.

## 4.4 ABLATION STUDY

We further verify CollaQ in the ablation study. First, we show that CollaQ outperforms a baseline (`SumTwoNets`) that simply sums over two networks which takes the agent's full observation as the input. `SumToNets` does not distinguish between $Q^{\mathrm{alone}}$ (which only takes $s_i$ as the input) and $Q^{\mathrm{collab}}$ (which respects the condition $Q^{\mathrm{collab}}(s_i, \cdot) = 0$). Second, we show that MARA loss is indeed critical for the performance of CollaQ.

We compare our method with `SumTwoNets` trained with QMIX in each agent. The baseline has a similar parameter size compared to CollaQ. As shown in Fig. 8, comparing to `SumTwoNets` trained with QMIX, CollaQ improves the win rates by $17\%$-$47\%$ on hard scenarios. We also study the importance of MARA Loss by removing it from CollaQ. Using MARA Loss boosts the performance by $14\%$-$39\%$ on hard scenarios, consistent with the decomposition proposed in Sec. 2.3.

## 5 RELATED WORK

Multi-agent reinforcement learning (MARL) has been studied since the 1990s (Tan, 1993; Littman, 1994; Bu et al., 2008). Recent progresses of deep reinforcement learning give rise to an increasing effort of designing general-purpose deep MARL algorithms (including COMA (Foerster et al., 2018), MADDPG (Lowe et al., 2017), MAPPO (Berner et al., 2019), PBT (Jaderberg et al., 2019), MAAC (Iqbal and Sha, 2018), etc) for complex multi-agent games. We utilize the Q-learning framework and consider the collaborative tasks in strategic games. Other works focus on different aspects of collaborative MARL setting, such as learning to communicate (Foerster et al., 2016; Sukhbaatar et al., 2016; Mordatch and Abbeel, 2018), robotics manipulation (Chitnis et al., 2019), traffic control (Vinitsky et al., 2018), social dilemmas (Leibo et al., 2017), etc.

The problem of ad hoc team play in multiagent cooperative games was raised in the early 2000s (Bowling and McCracken, 2005; Stone et al., 2010) and is mostly studied in the robotic soccer domain (Hausknecht et al., 2016). Most works (Barrett and Stone, 2015; Barrett et al., 2012; Chakraborty

and Stone, 2013; Woodward et al., 2019) either require sophisticated online learning at test time or require strong domain knowledge of possible teammates, which poses significant limitations when applied to complex real-world situations. In contrast, our framework achieves *zero-shot* generalization and requires little changes to the overall existing MARL training. There are also works considering a much simplified ad-hoc teamwork setting by tackling a varying number of test-time homogeneous agents (Schwab et al., 2018; Long et al., 2020) while our method can handle more general scenarios.

Previous work on the generalization/robustness in MARL typically considers a competitive setting and aims to learn policies that can generalize to different test-time *opponents*. Popular techniques include meta-learning for adaptation (Al-Shedivat et al., 2017), adversarial training (Li et al., 2019), Bayesian inference (He et al., 2016; Shen and How, 2019; Serrino et al., 2019), symmetry breaking (Hu et al., 2020), learning Nash equilibrium strategies (Lanctot et al., 2017; Brown and Sandholm, 2019) and population-based training (Vinyals et al., 2019; Long et al., 2020; Canaan et al., 2020). Population-based algorithms use ad hoc team play as a *training component* and the overall objective is to improve opponent generalization. Whereas, we consider zero-shot generalization to different teammates at *test time*. Our work is also related to the hierarchical approaches for multi-agent collaborative tasks (Shu and Tian, 2019; Carion et al., 2019; Yang et al., 2020). They train a centralized manager to assign subtasks to individual workers and it can generalize to new workers at test time. However, all these works assume known worker types or policies, which is infeasible for complex tasks. Our method does not make any of these assumptions and can be easily trained in an end-to-end fashion.

There have also been effort on decomposing the observation space through individual networks. ASN (Wang et al., 2019) decomposes the observation space of each agent trying to capture semantic meaning of actions, DyAN (Wang et al., 2020) adopts similar architecture in a curriculum domain. EPC (Long et al., 2020) also proposes to use attention between individual agents to make the network structure invariant to the size of agents. While the network structure of CollaQ to some extent share some similarity with the works aforementioned, the semantic meaning of each component is different. CollaQ models the interaction between agents using an alone network and an attention-based collaborative network, one used to model self-interest solutions and the other one models the influence of other agents on the particular agent.

Several papers also discuss social dilemma in a multi-agent setting (Leibo et al., 2017; Rapoport, 1974; Van Lange et al., 2013). Several works in reinforcement learning have been proposed to solve problems such as prisoner's delimma Sandholm and Crites (1996); de Cote et al. (2006); Wunder et al. (2010). However, in our setting, all the agents share the same environmental reward. Thus, the optimal solution for all the agents is to jointly optimize the shared reward. SSD Jaques et al. (2019) gives the agent an extra intrinsic reward when its action has huge influence on others. CollaQ does not use any intrinsic reward.

Lastly, our mathematical formulation is related to the credit assignment problem in RL (Sutton, 1985; Foerster et al., 2018; Nguyen et al., 2018). Some reward shaping literature also fall into this category (Devlin et al., 2014; Devlin and Kudenko, 2012). But our approach does not calculate any explicit reward assignment, we distill the theoretical insight and derive a simple yet effective learning objective.

## 6 CONCLUSION

In this work, we propose CollaQ that models Multi-Agent RL as a dynamic reward assignment problem. We show that under certain conditions, there exist decentralized policies for each agent *and* these policies are approximately optimal from the point of view of a team goal. CollaQ then learns these policies by resorting to an end-to-end training framework while using decomposition in $Q$-function suggested by the theoretical analysis. CollaQ is tested in a complex practical StarCraft MultiAgent Challenge and surpasses previous SoTA by $40\%$ in terms of win rates on various maps and $30\%$ in several ad hoc team play settings. We believe the idea of multi-agent reward assignment used in CollaQ can be an effective strategy for ad hoc MARL.

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

# A   COLLABORATIVE Q DETAILS

We derive the gradient and provide the training details for Eq. 5.

**Gradient for Training Objective**. Taking derivative w.r.t $\theta_n^a$ and $\theta_n^c$ in Eq. 5, we arrive at the following gradient:

$$\nabla_{\theta_n^a} \mathbf{L}_n(\theta_n^a, \theta_n^c) = \mathbb{E}_{s_i,a \sim \rho(\cdot),r_i;s' \sim \varepsilon}[(r + \gamma \max_{a'} Q_i(s', a', r_i; \theta_{n-1}^a, \theta_{n-1}^c) - Q_i(o_i, a, r_i; \theta_n^a, \theta_n^c))$$
$$\nabla_{\theta_n^a} Q_i^a(s_i, a, r_i; \theta_n^a)] \tag{6a}$$

$$\nabla_{\theta_n^c} \mathbf{L}_n(\theta_n^a, \theta_n^c) = \mathbb{E}_{s_i,a \sim \rho(\cdot),r_i;s' \sim \varepsilon}[(r + \gamma \max_{a'} Q_i(s', a', r_i; \theta_{n-1}^a, \theta_{n-1}^c) - Q_i(o_i, a, r_i; \theta_n^a, \theta_n^c))$$
$$\nabla_{\theta_n^c} Q_i^c(o_i, a, r_i; \theta_n^c) - \alpha Q_i^c(s_i, a, r_i; \theta_n^c) \nabla_{\theta_n^c} Q_i^c(s_i, a, r_i; \theta_n^c)] \tag{6b}$$

**Soft CollaQ**. In the actual implementation, we use a soft-constraint version of CollaQ: we subtract $Q^{\mathrm{collab}}(o_i^{alone}, a_i)$ from Eq. 4. The *Q-value Decomposition* now becomes:

$$Q_i(o_i, a_i) = Q_i^{\mathrm{alone}}(o_i^{alone}, a_i) + Q_i^{\mathrm{collab}}(o_i, a_i) - Q^{\mathrm{collab}}(o_i^{alone}, a_i) \tag{7}$$

The optimization objective is kept the same as in Eq. 5. This helps reduce variances in all the settings in resource collection and Starcraft multi-agent challenge. We sometimes also replace $Q^{\mathrm{collab}}(o_i^{alone}, a_i)$ in Eq. 7 by its target to further stabilize training.

# B   ENVIRONMENT SETUP AND TRAINING DETAILS

**Resource Collection**. We set the discount factor as 0.992 and use the RMSprop optimizer with a learning rate of 4e-5. $\epsilon$-greedy is used for exploration with $\epsilon$ annealed linearly from 1.0 to 0.01 in $100k$ steps. We use a batch size of 128 and update the target every 10k steps. For temperature parameter $\alpha$, we set it to 1. We run all the experiments for 3 times and plot the mean/std in all the figures.

**StarCraft Multi-Agent Challenge**. We set the discount factor as 0.99 and use the RMSprop optimizer with a learning rate of 5e-4. $\epsilon$-greedy is used for exploration with $\epsilon$ annealed linearly from 1.0 to 0.05 in $50k$ steps. We use a batch size of 32 and update the target every 200 episodes. For temperature parameter $\alpha$, we set it to 0.1 for 27m_vs_30m and to 1 for all other maps.

All experiments on StarCraft II use the default reward and observation settings of the SMAC benchmark. For ad hoc team play with different VIP, an additional 100 reward is added to the original 200 reward for winning the game if the VIP agent is alive after the episode.

For swapping agent types, we design the maps 3s1z_vs_16zg, 1s3z_vs_16zg and 2s2z_vs_16zg (**s** stands for stalker, **z** stands for zealot and **zg** stands for zergling). We use the first two maps for training and the third one for testing. For adding units, we use 27m_vs_30m for training and 28m_vs_30m for testing (**m** stands for marine). For removing units, we use 29m_vs_30m for training and 28m_vs_30m for testing.

We run all the experiments for 4 times and plot the mean/std in all the figures.

# C   DETAILED RESULTS FOR RESOURCE COLLECTION

We compare CollaQ with QMIX and CollaQ with attention-based model in resource collection setting. As shown in Fig. 9, QMIX does not show great performance as it is even worse than random action. Adding attention-based model introduces a larger variance, so the performance degrades by 10.66 in training but boosts by 2.13 in ad ad hoc team play.

# D   DETAILED RESULTS FOR STARCRAFT MULTI-AGENT CHALLENGE

We provide the win rates for CollaQ and QMIX on the environments without random agent IDs on three maps. Fig. 10 shows the results for both method.

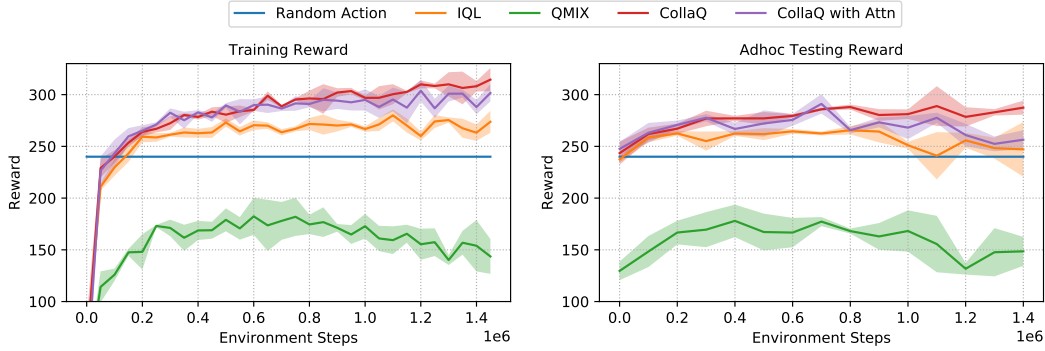

Figure 9: Results for resource collection. Adding attention-based model to CollaQ introduces a larger variance so the performance is a little worse. QMIX does not show good performance in this setting.

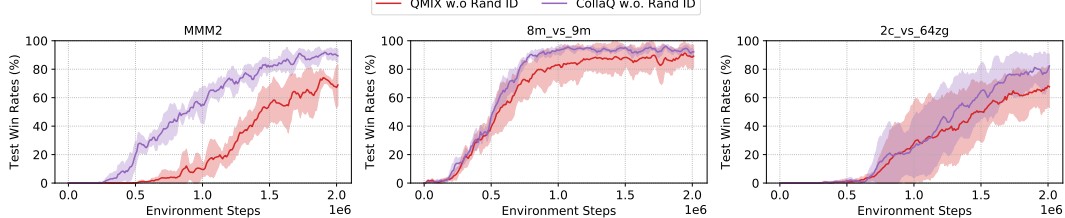

Figure 10: Results for StarCraft Multi-Agent Challenge without random agent IDs. CollaQ outperforms QMIX on all three maps.

We show the exact win rates for all the maps and settings mentioned in StarCraft Multi-Agent Challenge. From Tab. 1, we can clearly see that CollaQ improves the previous SoTA by a large margin.

Table 1: Win rates for StarCraft Multi-Agent Challenge. CollaQ show superior performance over all baselines.

|            | IQL     | VDN     | QTRAN   | QMIX    | CollaQ     | CollaQ with Attn |
|------------|---------|---------|---------|---------|------------|------------------|
| 5m_vs_6m   | 62.81%  | 69.37%  | 35.31%  | 66.25%  | **81.88**% | 80.00%           |
| MMM2       | 4.22%   | 6.41%   | 0.32%   | 36.56%  | 79.69%     | **84.69**%       |
| 2c_vs_64zg | 33.75%  | 22.66%  | 8.13%   | 34.06%  | **87.03**% | 62.66%           |
| 27m_vs_30m | 1.10%   | 6.88%   | 0.00%   | 19.06%  | 41.41%     | **50.63**%       |
| 8m_vs_9m   | 71.09%  | 82.66%  | 28.75%  | 77.97%  | 92.19%     | **96.41**%       |
| 10m_vs_11m | 70.47%  | 86.56%  | 31.10%  | 81.10%  | 91.25%     | **97.50**%       |

We also check the margin of winning scenarios, measured as how many units survive after winning the battle. The experiments are repeated over 128 random seeds. CollaQ surpasses the QMIX by over 2 units on average (Tab. 2), which is a huge gain.

Table 2: Number of survived units on six StaCraft maps. We compute mean and standard deviation over 128 runs. CollaQ outperforms all baselines significantly by managing more units to survive.

|                 | 5m_vs_6m         | MMM2             | 2c_vs_64zg       | 27m_vs_30m       | 8m_vs_9m         | 10m_vs_11m       |
|-----------------|------------------|------------------|------------------|------------------|------------------|------------------|
| IQL             | $0.91 \pm 0.28$  | $0.02 \pm 0.03$  | $0.05 \pm 0.04$  | $0.00 \pm 0.00$  | $0.95 \pm 0.36$  | $0.6 \pm 0.44$   |
| VDN             | $1.35 \pm 0.13$  | $0.28 \pm 0.32$  | $0.23 \pm 0.12$  | $0.55 \pm 0.93$  | $3.16 \pm 0.61$  | $3.39 \pm 1.44$  |
| QTRAN           | $1.76 \pm 0.53$  | $0.31 \pm 0.44$  | $0.36 \pm 0.35$  | $0.00 \pm 0.00$  | $2.43 \pm 0.53$  | $3.06 \pm 2.11$  |
| QMIX            | $1.72 \pm 0.5$   | $1.92 \pm 1.02$  | $0.47 \pm 0.11$  | $1.79 \pm 0.72$  | $2.75 \pm 0.48$  | $3.89 \pm 1.74$  |
| CollaQ          | $1.95 \pm 0.41$  | $\mathbf{4.89} \pm 1.32$ | $\mathbf{1.48} \pm 0.15$ | $2.80 \pm 0.94$  | $\mathbf{3.98} \pm 0.56$ | $\mathbf{4.91} \pm 1.48$ |
| CollaQ with Attn| $\mathbf{2.77} \pm 0.17$ | $4.73 \pm 1.08$  | $1.00 \pm 0.49$  | $\mathbf{5.22} \pm 1.79$ | $3.68 \pm 0.63$  | $4.73 \pm 0.41$  |

In a simple ad hoc team play setting, we assign a new VIP agent whose survival matters at test time. Results in Tab. 3 show that at test time, the VIP agent in CollaQ has substantial higher survival rate than QMIX.

Table 3: VIP agents survival rates for StarCraft Multi-Agent Challenge. CollaQ with attention surpasses QMIX by a large margin.

|  | IQL | VDN | QTRAN | QMIX | CollaQ | CollaQ with Attn |
|---|---|---|---|---|---|---|
| 5m_vs_6m | 30.47% | 46.72% | 16.72% | 38.13% | 56.72% | **61.72**% |
| MMM2 | 0.31% | 0.63% | 0.16% | 30.16% | 62.34% | **81.41**% |
| 8m_vs_9m | 37.35% | 47.34% | 6.25% | 48.91% | 59.06% | **78.13**% |

We also test CollaQ in a harder ad hoc team play setting: swapping/adding/removing agents at test time. Tab 4 summarizes the results for ad hoc team play, CollaQ outperforms QMIX by a lot.

Table 4: Win rates for StarCraft Multi-Agent Challenge with swapping/adding/removing agents. CollaQ improves QMIX substantially.

|  | IQL | VDN | QTRAN | QMIX | CollaQ | CollaQ with Attn |
|---|---|---|---|---|---|---|
| Swapping | 0.00% | 18.91% | 0.00% | 37.03% | 46.25% | **46.41**% |
| Adding* | 13.44% | 23.28% | 0.16% | 70.94% | - | **79.22**% |
| Removing* | 0.94% | 16.41% | 0.16% | 58.44% | - | **73.12**% |

* IQL, VDN, QTRAN and QMIX here all use attention-based models.

## E   VIDEOS AND VISUALIZATIONS OF STARCRAFT MULTI-AGENT CHALLENGE

We extract several video frames from the replays of CollaQ's agents for better visualization. In addition to that, we provide the full replays of QMIX and CollaQ. CollaQ's agents demonstrate super interesting behaviors such as healing the agents under attack, dragging back the unhealthy agents, and protecting the VIP agent (under the setting of ad hoc team play with different VIP agent settings). The visualizations and videos are available at `https://sites.google.com/view/collaq-starcraft`

## F   PROOF AND LEMMAS

**Lemma 1.** *If $a_1' \geq a_1$, then $0 \leq \max(a_1', a_2) - \max(a_1, a_2) \leq a_1' - a_1$.*

*Proof.* Note that $\max(a_1, a_2) = \frac{a_1 + a_2}{2} + \left| \frac{a_1 - a_2}{2} \right|$. So we have:

$$\max(a_1', a_2) - \max(a_1, a_2) = \frac{a_1' - a_1}{2} + \left| \frac{a_1' - a_2}{2} \right| - \left| \frac{a_1 - a_2}{2} \right| \leq \frac{a_1' - a_1}{2} + \left| \frac{a_1 - a_1'}{2} \right| = a_1' - a_1 \tag{8}$$

□

### F.1   LEMMAS

**Lemma 2.** *For a Markov Decision Process with finite horizon $H$ and discount factor $\gamma < 1$. For all $i \in \{1, \ldots, K\}$, all $\mathbf{r}_1, \mathbf{r}_2 \in \mathbb{R}^M$, all $s_i \in S_i$, we have:*

$$|V_i(s_i; \mathbf{r}_1) - V_i(s_i; \mathbf{r}_2)| \leq \sum_{x,a} \gamma^{|s_i - x|} |r_1(x, a) - r_2(x, a)| \tag{9}$$

*where $|s_i - x|$ is the number of steps needed to move from $s_i$ to $x$.*

*Proof.* By definition of optimal value function $V_i$ for agent $i$, we know it satisfies the following Bellman equation:

$$V_i(x_h; \mathbf{r}_i) = \max_{a_i} \left( r_i(x_i, a_i) + \gamma \mathbb{E}_{x_{h+1} | x_h, a_h} [V_i(x_{h+1})] \right) \tag{10}$$

Note that to avoid confusion between agents initial states $\mathbf{s} = \{s_1, \ldots, s_K\}$ and reward at state-action pair $(s, a)$, we use $(x, a)$ instead. For terminal node $x_H$, which exists due to finite-horizon MDP with horizon $H$, $V_i(x_H) = r_i(x_H)$. The current state $s_i$ is at step 0 (i.e., $x_0 = s_i$).

We first consider the case that $\mathbf{r}_1$ and $\mathbf{r}_2$ only differ at a single state-action pair $(x_h^0, a_h^0)$ for $h \leq H$. Without loss of generality, we set $r_1(x_h^0, a_h^0) > r_2(x_h^0, a_h^0)$.

By definition of finite horizon MDP, $V_i(x_{h'}; \mathbf{r}_1) = V_i(x_{h'}; \mathbf{r}_2)$ for $h' > h$. By the property of max function (Lemma 1), we have:

$$0 \leq V_i(x_h^0; \mathbf{r}_1) - V_i(x_h^0; \mathbf{r}_2) \leq r_1(x_h^0, a_h^0) - r_2(x_h^0, a_h^0) \tag{11}$$

Since $p(x_h^0 | x_{h-1}, a_{h-1}) \leq 1$, for any $(x_{h-1}, a_{h-1})$ at step $h - 1$, we have:

$$0 \leq \gamma \left[ \mathbb{E}_{x_h | x_{h-1}, a_{h-1}} [V_i(x_h; \mathbf{r}_1)] - \mathbb{E}_{x_h | x_{h-1}, a_{h-1}} [V_i(x_h; \mathbf{r}_2)] \right] \tag{12}$$

$$\leq \gamma \left[ r_1(x_h^0, a_h^0) - r_2(x_h^0, a_h^0) \right] \tag{13}$$

Applying Lemma 1 and notice that all other rewards do not change, we have:

$$0 \leq V_i(x_{h-1}; \mathbf{r}_1) - V_i(x_{h-1}; \mathbf{r}_2) \leq \gamma \left[ r_1(x_h^0, a_h^0) - r_2(x_h^0, a_h^0) \right] \tag{14}$$

We do this iteratively, and finally we have:

$$0 \leq V_i(s_i; \mathbf{r}_1) - V_i(s_i; \mathbf{r}_2) \leq \gamma^h \left[ r_1(x_h^0, a_h^0) - r_2(x_h^0, a_h^0) \right] \tag{15}$$

We could show similar case when $r_1(x_h^0, a_h^0) < r_2(x_h^0, a_h^0)$, therefore, we have:

$$|V_i(s_i; \mathbf{r}_1) - V_i(s_i; \mathbf{r}_2)| \leq \gamma^h |r_1(x_h^0, a_h^0) - r_2(x_h^0, a_h^0)| \tag{16}$$

where $h = |x_h^0 - s_i|$ is the distance between $s_i$ and $x_h^0$.

Now we consider general $\mathbf{r}_1 \neq \mathbf{r}_2$. We could design path $\{\mathbf{r}_t\}$ from $\mathbf{r}_1$ to $\mathbf{r}_2$ so that each time we only change one distinct reward entry. Therefore each $(s, a)$ pairs happens only at most once and we have:

$$|V_i(s_i; \mathbf{r}_1) - V_i(s_i; \mathbf{r}_2)| \leq \sum_t |V_i(s_i; \mathbf{r}_{t-1}) - V_i(s_i; \mathbf{r}_t)| \tag{17}$$

$$\leq \sum_{x, a} \gamma^{|x - s_i|} |r_1(x, a) - r_2(x, a)| \tag{18}$$

$\square$

## F.2 Thm. 1

First we prove the following lemma:

**Lemma 3.** *For any reward assignments $\mathbf{r}_i$ for agent $i$ for the optimization problem (Eqn. 1) and a local reward set $M_i^{\mathrm{local}} \supseteq \{x : |x - s_i| \leq C\}$, if we construct $\tilde{\mathbf{r}}_i$ as follows:*

$$\tilde{r}_i(x, a) = \begin{cases} r_i(x, a) & x \in M_i^{\mathrm{local}} \\ 0 & x \notin M_i^{\mathrm{local}} \end{cases} \tag{19}$$

*Then we have:*

$$|V_i(s_i; \mathbf{r}_i) - V_i(s_i; \tilde{\mathbf{r}}_i)| \leq \gamma^C R_{\max} M \tag{20}$$

*where $M$ is the total number of sparse reward sites and $R_{\max}$ is the maximal reward that could be assigned at each reward site $x$ while satisfying the constraint $\phi(r_1(x, a), r_2(x, a), \ldots, r_K(s, a)) \leq 0$.*

*Proof.* By Lemma 2, we know that

$$|V_i(s_i; \mathbf{r}_i^*) - V_i(s_i; \tilde{\mathbf{r}}_i)| \leq \sum_{x \notin S_i^{\mathrm{local}}} \gamma^{|x - s_i|} |r_i^*(s, a) - \tilde{r}_i(s, a)| \tag{21}$$

$$\leq \gamma^C \sum_{x \notin S_i^{\mathrm{local}}} |r_i^*(s, a)| \tag{22}$$

$$\leq \gamma^C R_{\max} M \tag{23}$$

$\square$

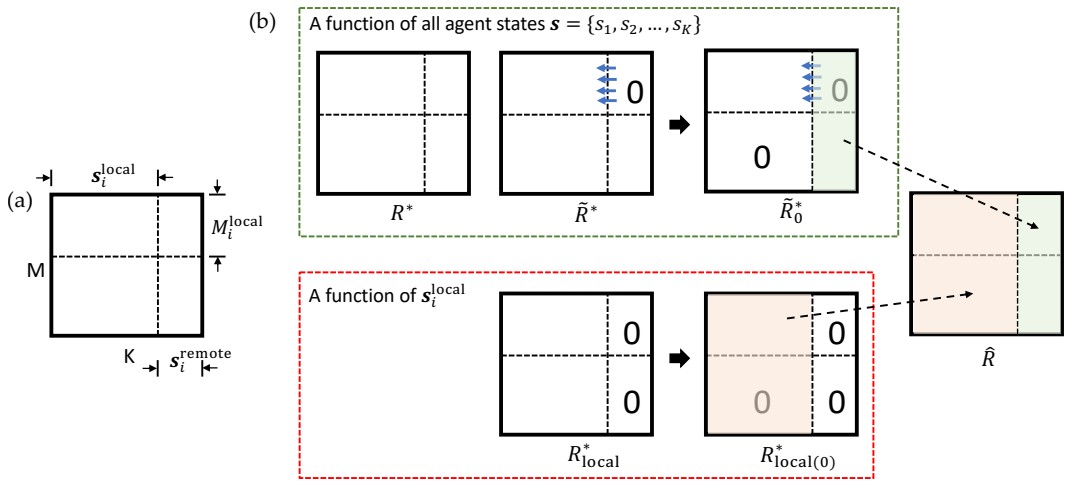

Figure 11: Different reward assignments.

Note that "sparse reward site" is important here, otherwise there could be exponential sites $x \notin S_i^{\text{local}}$ and Eqn. 23 becomes vacant.

Then we prove the theorem.

*Proof.* Given a constant C, for each agent $i$, we define the vicinity reward site $B_i(C) := \{x : |x - s_i| \leq C\}$.

Given agent $i$ and its local "buddies" $\mathbf{s}_i^{\text{local}}$ (a subset of multiple agent indices), we construct the corresponding reward site set $M_i^{\text{local}}$:

$$M_i^{\text{local}} = \bigcup_{s_j \in \mathbf{s}_i^{\text{local}}} B_j(C) \tag{24}$$

Define the remote agents $\mathbf{s}_i^{\text{remote}} = \mathbf{s} \backslash \mathbf{s}_i^{\text{local}}$ as all agents that do not belong to $\mathbf{s}_i^{\text{local}}$.

Define the distance $D$ between the $M_i^{\text{local}}$ and $\mathbf{s}_i^{\text{remote}}$:

$$D = \min_{x \in M_i^{\text{local}}} \min_{s_j \in \mathbf{s}_i^{\text{remote}}} |x - s_j| \tag{25}$$

Intuitively, the larger $D$ is, the larger distance between relevant rewards sites from remote agents and the tighter the bound. There is a trade-off between $C$ and $D$: the larger the vicinity, $M_i^{\text{local}}$ expands and the smaller $D$ is.

Given this setting, we then construct a few reward assignments (see Fig. 11), given the current agent states $\mathbf{s} = \{s_1, s_2, \ldots, s_K\}$. For brevity, we write $R[M, \mathbf{s}]$ to be the submatrix that relates to reward site $M$ and agents set $\mathbf{s}$.

- The optimal solution $R^*$ for Eqn. 1.

- The perturbed optimal solution $\tilde{R}^*$ by pushing the reward assignment of $[M_i^{\text{local}}, \mathbf{s}_i^{\text{remote}}]$ in $R^*$ to $[M_i^{\text{local}}, \mathbf{s}_i^{\text{local}}]$.

- From $\tilde{R}^*$, we get $\tilde{R}_0^*$ by setting the region $[M_i^{\text{remote}}, \mathbf{s}_i^{\text{local}}]$ to be zero.

- The local optimal solution $R_{\text{local}}^*$ that only depends on $\mathbf{s}_i^{\text{local}}$. This solution is obtained by setting $[:, \mathbf{s}_i^{\text{remote}}]$ to be zero and optimize Eqn. 1.

- From $R_{\text{local}}^*$, we get $R_{\text{local}(0)}^*$ by setting $[M_i^{\text{remote}}, \mathbf{s}_i^{\text{local}}]$ to be zero.

It is easy to show all these rewards assignment are feasible solutions to Eqn. 1. This is because if the original solution is feasible, then setting some reward assignment to be zero also yields a feasible solution, due to the property of the constraint $\phi$.

For simplicity, we define $J_{\text{local}}$ to be the partial objective that sums over $s_j \in \mathbf{s}_i^{\text{local}}$ and similarly for $J_{\text{remote}}$.

We could show the following relationship between these solutions:

$$J_{\text{remote}}(\tilde{R}^*) \geq J_{\text{remote}}(R^*) - \gamma^D R_{\max} MK \tag{26}$$

This is because each of this reward assignment move costs at most $\gamma^D R_{\max}$ by Lemma 2 and there are at most $MK$ such movement.

On the other hand, for each $s_j \in \mathbf{s}_j^{\text{local}}$, since $M_i^{\text{local}} \supseteq B_j(C)$, from Lemma 3 we have:

$$V_j(R^*_{\text{local}(0)}) \geq V_j(R^*_{\text{local}}) - \gamma^C R_{\max} M \tag{27}$$

And similarly we have:

$$V_j(\tilde{R}^*_0) \geq V_j(\tilde{R}^*) - \gamma^C R_{\max} M \tag{28}$$

Now we construct a new solution $\hat{R}_i$ by combining $R^*_{\text{local}(0)}[:, \mathbf{s}_i^{\text{local}}]$ with $\tilde{\mathbf{r}}_0^*[:, \mathbf{s}_i^{\text{remote}}]$. This is still a feasible solution since in both $R^*_{\text{local}(0)}$ and $\tilde{R}_0^*$, their top-right and bottom-left sub-matrices are zero, and its objective is still good:

$$J(\hat{R}) = J_{\text{local}}(R^*_{\text{local}(0)}) + J_{\text{remote}}(\tilde{R}_0^*) \tag{29}$$

$$\overset{①}{\geq} J_{\text{local}}(R^*_{\text{local}}) - \gamma^C R_{\max} MK + J_{\text{remote}}(\tilde{R}_0^*) \tag{30}$$

$$\overset{②}{\geq} J_{\text{local}}(\tilde{R}^*) + J_{\text{remote}}(\tilde{R}_0^*) - \gamma^C R_{\max} MK \tag{31}$$

$$\overset{③}{\geq} J_{\text{local}}(R^*) + J_{\text{remote}}(\tilde{R}_0^*) - \gamma^C R_{\max} MK \tag{32}$$

$$\overset{④}{=} J_{\text{local}}(R^*) + J_{\text{remote}}(\tilde{R}^*) - \gamma^C R_{\max} MK \tag{33}$$

$$\overset{⑤}{\geq} J_{\text{local}}(R^*) + J_{\text{remote}}(R^*) - R_{\max} MK(\gamma^C + \gamma^D) \tag{34}$$

$$\overset{⑥}{=} J(R^*) - R_{\max} MK(\gamma^C + \gamma^D) \tag{35}$$

Note that ① is due to Eqn. 27, ② is due to the optimality of $R^*_{\text{local}}$ (and looser constraints for $R^*_{\text{local}}$), ③ is due to the fact that $\tilde{R}^*$ is obtained by *adding* rewards released from $\mathbf{s}_i^{\text{remote}}$ to $\mathbf{s}_i^{\text{local}}$. ④ is due to the fact that $\tilde{R}_0^*$ and $\tilde{R}^*$ has the same remote components. ⑤ is due to Eqn. 26. ⑥ is by definition of $J_{\text{local}}$ and $J_{\text{remote}}$.

Therefore we obtain $\hat{\mathbf{r}}_i = [\hat{R}]_i$ that only depends on $\mathbf{s}_i^{\text{local}}$. On the other hand, the solution $\hat{R}$ is close to optimal $R^*$, with gap $(\gamma^C + \gamma^D)R_{\max} MK$. $\qquad\square$

