# OpenReview forum: "Multi-Agent Collaboration via Reward Attribution Decomposition"
_ICLR.cc/2021/Conference — Reject_

### Official Review · AnonReviewer4 · 2020-10-17
**Nice idea, but needs some serious clarification and polishing**

**Rating:** 5
**Confidence:** 2

**Review:**

The paper studies a team of agents that collaborate to maximize a global objective, where all agents receive the same reward value as a function of their state and actions. The approach suggested here is to assign each agent a virtual ("perceived") reward function such that all the virtual rewards sum up to the actual reward. Then the problem from the point of view of agent i can be solved with local value functions that depend on the local state of agent i and its perceived reward.  It is shown that such a decoupled policy exists that approximates the optimal policy if the state structure decomposes "well enough" into local states. Experiments demonstrate the advantages of this approach on a resource collection game on the StarCraft Multi-Agent Challange, with dramatic improvements over existing methods for the scenarios that were tested.

The paper is mostly easy to follow, and is well-written and well motivated. The idea of this paper is nice and intuitive. Being less familiar with the literature, It's hard for me to judge how novel this idea is. It seems like there must exist multi-agent works that exploit the local structure of the interaction to reduce the dimensionality. Hence, as a start, the contribution of this paper can be enhanced by extending the literature review to include works of that kind, even if under different contexts. If indeed this idea is unique enough (and had a major impact on the design of the algorithm), then this is a significant contribution. Then, the other main issue is with improving the rigor of the presentation and the math. While it's clear what the results are showing, some guessing is needed to fill in some gaps.

The reward and state structure: I guess that the "external rewards" referred to above (1) and the "same reward" that all agents share are the same thing. Please clarify and unify the definitions. Then it's not clear what's going on with the dimensions of the local states. The local states appear in the second paragraph of Section 2 and are never properly defined, and also the local state spaces S_i are not well defined. Are the cardinalities of S_i and A_i the same for all i? otherwise, the constraint in (1) isn't clear. Additionally, it's not clear how can one measure distances of states that live in different state spaces (of different agents) as is often being done in the proofs (e.g., the proof of Theorem 1, the definition of D, and so on). On that note, the idea of "nearby agents" should be more rigorously defined. Ignoring these gaps, it looks like Theorem 1 basically shows that the better the problem decomposes into local environments, the easier it becomes to solve it distributedly. It's not clear if the math provides any added value on top of this important yet simple observation.

The connection between theory and practice: Assigning perceived rewards to simplify the MARL problem is an elegant and appealing idea. For this reason, it's important to carefully discuss to what extent this idea actually influenced the algorithm that was tested in practice. Subsection 2.3 raises some concerns in this regard since the algorithm resorts to "end-to-end learning of Q_i", in what seems like a total bypass of the idea of the perceived rewards. Looking at equation (3) or (4), one can get the impression that the perceived rewards are just an interpretation of what happens "inside" when training the Q-values after splitting them as in (4). By itself, (4) makes a lot of sense and is very natural, so it's not clear if it isn't' easier to come up directly with (4) without knowing anything about perceived rewards. This raises the question: why isn't it possible to bypass the idea of the perceived rewards and motivate the paper based on (4), which is closer to the practical algorithm? answering this question is crucial to claim a significant contribution since otherwise the are two loosely related parts in this paper.

Precise statements: The statements of the mathematical results often lack some definitions. The statement of theorem 1 doesn't define R_{max} (but Lemma 3 does). The statement of Lemma 3 doesn't define the distance between states (but Lemma 2 does). Please make the statements more standalone and well defined. On the same note, make sure that important definitions don't randomly appear in the paper, sometimes too late (e.g., local states).

Experiments: The experimental results are overall nice and promising.  My only question here is why the resource collection scenario only compares to IQL and not to QTRAN/VDN/QMIX like the StarCraft scenario?

Vague sentences and typos:

"each agent i is acted independently on its own state" - acting? based on its own state?

(s_1,...,s_N) (bottom of page 2) - needs to be s_K

"so that" in Theorem 1 - such that

"We found that using the observation o_i of agent i covers s_i^{local} works sufficiently well" - not clear.

"since 1990s" - since the 1990s.

"We sometimes also replace... in Eq.7 by its target to further stabilize training" (Page 12)- what does sometimes mean? how can one reproduce this?

"doesn't" - does not

"Applying Lemma 1 and notice that all other rewards does not change" - do not change

"Define the remote agents s_i^{remote}..." isn't that a set of states and not of agents? please rephrase.

"the more distant between relevant rewards istes from remote agents" - the larger the distance?

---

> ### Author Response · Authors · 2020-11-13
> **Author Response to R4**
>
> We thank R4 for all the comments. Please also refer to the common questions above for the answer to the remaining questions
>
> **Q1**: How to measure the distance of states and "nearby agents" should be more rigorously defined.
>
> **A**: Please see the common questions part.
>
> **Q2**: What’s the meaning of math? Is there any implication beyond that the more the problem decomposes into local environments, the easier it becomes to solve it distributedly?
>
> **A**: With theorem 1, we can also see that it is valid to decompose the agent’s value function into $Q^{alone}$ and $Q^{collab}$. Thus we can derive our algorithm CollaQ with this important intuition. Please also check the common questions.
>
> **Q3**: Theorem 1 doesn't define $R_{max}$ and Lemma 3 doesn't define the distance between states. Please clarify notations.
>
> **A**: We thank the reviewers for pointing out. We’ll fix them in the next revision of the paper.
>
> **Q4**: Why in the resource collection scenario, CollaQ only compares to IQL and not to QTRAN/VDN/QMIX?
>
> **A**: There are two reasons: 1. Since the state of resource collection is fully observable, so IQL is enough for the agent to choose action. 2. We also provide results using QMIX on resource collection in Appendix Fig.9. QMIX doesn’t work well compared to IQL.

---

> > ### Comment · AnonReviewer4 · 2020-11-20
> > **The connection of the reward assignment aspect to the actual algorithm is still questionable, and a bit misleading in terms of the claimed novelty**
> >
> > Thanks for your response. While it answers some of my questions, the major concern remains.
> >
> > It is not very encouraging if the reward assignment approach was too complicated to conduct even a simple experiment based on it.
> >
> > I also have to disagree with the claim that the reward assignment view is necessary or a natural way to reach the actual algorithm that was tested and is showed in (4). This is of course subjective, which is exactly the problem - one would expect a stronger logical, algorithmic or mathematical connection between the story + theoretical idea and the algorithm + empirical results in this paper.
> >
> > It seems like (4) is very natural because if the problem decomposes well into local effects, it makes sense that part of the Q value would take only the individual/local state of the agent as in input. Isn't this the case? Am I missing something? this is the idea behind graphical games for example. Why is the reward assignment an easier way to motivate this? (e.g., isn't the MARA loss motivated enough by eliminating the ambiguity and to make the solution unique?).  I'd appreciate it if a more convincing (and less subjective) argument can be provided here, and in the paper as well.
> >
> > This issue is important both to make the paper more solid but also to establish novelty. I also haven't seen a multi-agent collaboration that uses reward assignment, but I also didn't see much of it here. Instead, there is a nice effort to "decompose the observation space" when it's possible, but as you discuss above, there are already works that presented similar approaches. The approach here is still novel, but not as innovative as the reward assignment makes it look like at first glance. Since the reward assignment is not really used here in practice, this is a bit misleading when trying to evaluate the novelty.
> >
> > I still didn't understand if all the dimensions of the local state space have to be equal, or how else can the constraint in (1) make sense. It seems like a restrictive assumption since different agents will have more or less integrated or isolated local environments.

---

> > > ### Author Response · Authors · 2020-11-24
> > > **Author Response for R4**
> > >
> > > We thank the reviewer for the comments.
> > >
> > > **Q1**: We think that the intuitive explanation that MARA loss is to eliminate the ambiguity and to make the solution unique is perfectly valid. However, we would like to emphasize several things here:
> > > 1. While (4) looks like a “natural” formulation, there are a lot of other natural formulations available (e.g., how about we have a $Q^{alone}$ function that takes as much input as possible? Instead of adding $Q^{alone}$ and $Q^{collab}$, why not multiple them together?). The theory helps us understand the principle behind it and nails down the precise formulation we use here.
> > > 2. Without the theory, there could be other possible decompositions/explanations of the algorithm. The theory provides one possible intuition/explanation behind CollaQ. The path to the solution, in addition to the solution itself, could trigger some interesting future research as well and should be regarded as a contribution of the paper. This is also recognized by other reviewers (e.g., R1: “Good theoretical analysis and compliant experiment performance”). For example, while currently, we don’t solve Eqn. 1 directly, the formulation itself could trigger other thoughts and connections to other existing works (e.g., multi-agent centralized planning).
> > > 3. The semantic meaning and intuition of our algorithm CollaQ are very different from previous work as we stated in the common questions.
> > >
> > > We have already added one section to the paper explaining the intuition and connection. We plan to leave a more in-depth analysis of the two to future work.
> > >
> > > Note that reward decomposition theory is only one part of our contributions. Our practical algorithm (motivated by intuitions and theory) CollaQ achieves impressive empirical performance not only on traditional multi-agent games like StarCraft II, but also outperforms all baselines by a good margin in our newly proposed multi-agent ad hoc setting. We think that those two parts should also be considered when we are trying to evaluate the novelty/contribution of the paper. Note that this is also mentioned by other reviewers (e.g., R3: “To the best of my knowledge, the MARA regularization is novel enough. The Ad-hoc MARL is an important problem in real-world applications but has not been fully studied.”, R2: “the main advance is in terms of scaling up to real-world scenarios rather than having a better explicit algorithm to manage reward decomposition at runtime in an online learning setting.”).
> > >
> > >
> > > **Q2**: We don’t require the dimension of local state $S^{local}$ to be the same for every agent. We use transformer architecture to put attention on the important part from observation $o_{i}$ that influences the agent’s value function.

---

### Official Review · AnonReviewer3 · 2020-10-21
**Reviews**

**Rating:** 6
**Confidence:** 3

**Review:**

To address the ad hoc team play, the authors propose a residual term of Q function, which additionally considers the states of nearby agents. A novel MARA loss is introduced to the residual term as a regularization to achieve the reward assignment implicitly. The proposed CollaQ could be easily built on QMIX and trained end-to-end. CollaQ outperforms other baselines on various tasks with the ad hoc team play setting.

The paper is very clear and well-structured. To the best of my knowledge, the MARA regularization is novel enough. The Ad-hoc MARL is an important problem in real-world applications but has not been fully studied. The interactive term with regularization is a practical and promising method to solve this problem and could be followed by other researchers.

However, I still have some concerns:

First, the theoretical analysis of reward assignment is not close to the implementation of CollaQ. There is no real assignment mechanism. A MARA loss is derived from the theoretical analysis to achieve the reward assignment implicitly, but the MARA loss could be more straightforwardly interpreted as that the Q value should be equal to the individual value when the agent cannot observe other agents. From this perspective, the complex reward assignment is not necessary. Moreover, CollaQ is built on QMIX. However, the individual value function in QMIX does not estimate a real expected return, and the value has no meaning. Is the theoretical analysis of reward assignment still valid in QMIX?

I do not find any experiments to support the claim that "agents using CollaQ would first learn to solve the problem pretending no other agents are around using Qalone then try to learn interaction with local agents through Qcollab." I think it is over-claimed and should be removed. Splitting the end-to-end learning process into two learning stages might harm the learning.

The visualizations in Fig 3 are helpful to understand how CollaQ works, but they are special cases. Statistical results are more convincing to verify how CollaQ influences the decision.

At the test time of StarCraft, are the IDs shuffled at each timestep or only at the first timestep of an episode?

----------Update after author response----------

I thank the authors for the detailed response. Most of my concerns have been addressed, and I decide to keep my score.

---

> ### Author Response · Authors · 2020-11-13
> **Author Response to R3**
>
> We thank R3 for all the comments. Please also refer to the common questions above for the answer to the remaining questions
>
> **Q1**: Do we still need the complex reward assignment if MARA loss could be more straightforwardly interpreted as that the Q value should be equal to the individual value when the agent cannot observe other agents?
>
> **A**: This is a good point. Your explanation is perfectly valid. In fact, this is actually the intuition behind the reward assignment and the Theorem just conveys this idea in a more formal way, so that it can be developed further (e.g., why we want to decompose this way). Please also see the connection between theory and the algorithm part in common questions.
>
> **Q2**: Is theoretical analysis still valid combined with QMIX?
>
> **A**: QMIX takes each individual Q function as the input. The decompositional property of each Q is not affected by how it is being used on the top level.
>
> **Q3**: Do agents using CollaQ would first learn to solve the problem pretending no other agents are around using $Q^{alone}$ then try to learn interaction with local agents through $Q^{collab}$?
>
> **A**: We thank the reviewer for pointing this out. We will revise this sentence in the next revision.
>
> **Q4**: The visualization of Fig3. is a special case. Statistical results would be preferred.
>
> **A**: We would like to emphasize that Fig. 3 is not a special case (it is actually randomly sampled) and this phenomenon happens quite often in our observation. We thank the reviewer for this suggestion and could report some statistics in the next revision of the paper. The visualizations combined with Theorem 1 and Eq. 3 could to some extent make the point.
>
> **Q5**: Are the IDs shuffled at each timestep or only at the first time step of an episode?
>
> **A**: The agent IDs are shuffled only at the first time step of an episode. In another word, each episode has a different (but fixed) ID assignment.

---

### Official Review · AnonReviewer2 · 2020-10-28
**Novel method for reward attribution achieves impressive results in state-of-the-art benchmark domains**

**Rating:** 7
**Confidence:** 3

**Review:**

The paper focuses on reward attribution in multiagent reinforcement learning, proposing a new algorithm, essentially by splitting value functions to what agents can achieve individually and learning separately how different individual rewards interact. This is an old idea, but the paper essentially applies it to the state-of-the-art deep learning machinery, producing impressive results on the hardest games state-of-the-art algorithms can manage.

As is the case with many similar areas, the work emphasises translation of an idea to the deelp learning setting, with the usual caveats, i.e. that there is not a major new methodological or conceptual insight, and the main advance is in terms of scaling up to real-world scenarios rather than having a better explicit algorithm to manage reward decomposition at runtime in an online learning setting. In other words, we learn more about how to solve challenging games rather than about the key underlying AI problem.

The paper does not offer a huge amount of novelty, but rather presents a solid deep RL engineering approach to solving the wider problem in a specific setting. Nonetheless, the technical material is well-developed, the presentation is overall of a high quality, and the experimental results extensive (and impressive).

---

> ### Author Response · Authors · 2020-11-13
> **Author Response to R2**
>
> We thank R2 for all the comments. Please also refer to the common questions above for the answer to the remaining questions
>
> **Q1**: There is not a major new methodological or conceptual insight.
>
> **A**: Please see the contribution part of how we view this work.

---

### Official Review · AnonReviewer1 · 2020-10-30
**Good theoretical analysis and compliant experiment performance**

**Rating:** 6
**Confidence:** 4

**Review:**

Good theoretical analysis and compliant experiment performance

1. The Limitation of Theorem 1: the authors have said that ``the optimal gap of r_i heavily depends on the size of s_i^{local}. But in experiments (including experiments in appendix), the authors only discussed the claimed optimal setting (“using the observation o_i of agent i covers $s_i^{local}$”). More experiments on other size of s_i^{local} could be added to better prove the conclusion. (Whether the best choice can not or hard to be proven mathematically.)
2. Some expression problems which may cause confusions: 1) Too many kinds of rewards including perceived reward, local reward, external reward and etc. The definitions of them are not very clear.  For example,  the perceived reward is really confused; 2) A brief algorithm flow chat and pseudo codes are needed for better understanding of how the algorithm works.
3. As this work is eventually an MARL work in solving ad hoc team setting games by decomposing reward. Some explicit comparisons (May be in form or experiments or brief analysis) should be added with some MARL methods(SSD: Social Influnce as Intrinsic Motivation for Multi-Agent Deep Reinforcement Learning; PBRS: Reward shaping for knowledge-based multi-objective multi-agent reinforcement learning). Only the credit assignment problem in RL is discussed, the authors need to discuss more on some other related works like social dilemma in MARL or reward shaping?

---

> ### Author Response · Authors · 2020-11-13
> **Author Response to R1**
>
> We thank R1 for all the comments. Please also refer to the common questions above for the answer to the remaining questions
>
> **Q1**: In Theorem 1, does observation cover $S^{i}_{local}$ a limitation?
>
> **A**: We thank the reviewer for this valid concern. This is actually one of the limitations of Theorem 1. However, we found that CollaQ works well empirically even in complex games like StarCraft II. We leave this limitation to future work.
>
> **Q2**: There are too many kinds of rewards including perceived reward, local reward, external reward.
>
> **A**: Please see the common questions section.
>
> **Q3**: Can you provide a brief algorithm flowchart and pseudo codes?
>
> **A**: The actual algorithm is simple so we didn’t draw an algorithm flowchart. The DQN training involves sampling episodes from a trajectory buffer and Bellman update using those samples. We adopt the DQN training paradigm (QMIX head on top) with the objective function defined in Eq. 5.

---

### Author Response · Authors · 2020-11-13
**Author Response to Common Questions (1/2)**

We are thankful for the reviewers’ insightful comments. We address common concerns here and will reply to each reviewer separately for their specific comments.

**Common Questions**:

**Contribution.**

[Some summarization of our work here.]

1. CollaQ proposes to formulate the multi-agent collaboration problem as a two-stage optimization.

2. Derived from the theorem, we present a practical algorithm that achieves the SoTA performance in complex games like StarCraft II.

3. We also try to solve ad hoc team play in a complicated game domain with our new framework and algorithm above. Empirical results of CollaQ on StarCraft II show good performance over the current algorithms.

4. To the best of our knowledge, we are the first to study ad hoc team play in a complex game domain without requiring sophisticated online learning at test time or strong domain knowledge of possible teammates.

**[R4]** To our best knowledge, we haven’t seen a similar formulation for multi-agent collaboration that uses the reward assignment to model the collaborative behavior of agents in prior works and believe this is a substantial contribution to the community.

**[R2]** We would like to emphasize that: 1. Our theorem on formulating the multi-agent ad hoc team play as a two-stage is by itself novel and could trigger future research. 2. Motivated by the theorem, the algorithm CollaQ (with MARA loss) is also novel and effective in realistic environments. 3. We try to solve ad hoc team play in a more complicated game domain with the novel framework and algorithm above. The experiments achieved strong results.

**What’s the connection between the theory and practical algorithm (CollaQ) [R2, R4]?**

In summary, we use the theory as motivation and derive a general end-to-end algorithm that can work empirically in complex/realistic environments. Besides motivating the practical algorithm, the theorem shows a bigger picture of the framework and can serve as a separate contribution to trigger future research along the direction. However, as mentioned in Sec 2.1 and 2.2, the optimization for reward assignment is complex and cannot be done efficiently. How to conduct such optimization efficiently in complex environments is beyond the scope of this paper. Thus, we didn’t conduct experiments on explicitly doing optimization on reward assignment and leave this for future work. Instead, derived from the theorem, CollaQ uses a theory-inspired Q decomposition and serves as a practical algorithm that can work in realistic environments.

Without the theory, if we just start from Eq. 4, then we will

1. Completely lose the motivation why we have these two specific decompositions. Why is $Q^{alone}$ only allowed to take the individual state of agent $i$ as the input? Why does $Q^{collab}$ need to have MARA loss? These questions are well-motivated by Eq. 3 and the theorem.

2. Lose a potentially bigger picture of reward assignment underlying Eqn. 4. The reward assignment formulation for multi-agent collaboration (Eqn. 1) could potentially trigger future research, even if it cannot be solved efficiently for now.

**Related Works [R1, R4]**

We thank the reviewers for the related works. We’ll definitely update the paper accordingly in the next revision. We just want to clarify the difference between CollaQ and certain related works on decomposition network, social influence, reward shaping:

Decomposition Network: There is other literature that tries to decompose the observation space: ASN [2] decomposes the observation space of each agent trying to capture semantic meaning of actions, DyAN [1] adopts similar architecture in a curriculum domain. We would like to point out that although the network structure shares some similarities, the semantic meaning of each component is different. CollaQ is well-motivated by Theorem 1 and the derived MARA loss is also novel. Thus the algorithm CollaQ shares little similarity except for the network structure with the papers mentioned before. Fig. 3 also supports this idea.

Social Influence: the SSD [3] paper gives the agent an extra intrinsic reward when its action has huge influence on others. In contrast, CollaQ doesn’t need any intrinsic reward for learning collaboration behaviors. In addition to that, CollaQ doesn’t need extra monte-carlo simulation in the environments, which is not possible in games like StarCraft II. This simulation is needed for SSD to find the most influential action of an agent.

Social dilemma [5]: social dilemma often assumes each agent has their independent reward, but in our setting, the agents are fully cooperative and share the same reward.

Reward shaping [4]: Although our theorem is motivated by reward assignment, our practical algorithm CollaQ doesn’t do reward shaping explicitly. The only reward it receives is the external reward (e.g., for SMAC, whether the team wins against the opponent team).

(to be continued)

---

### Author Response · Authors · 2020-11-13
**Author Response to Common Questions (2/2)**


**What’s the Reward structure and how to define $S^{i}_{local}$? [R1, R4]**

External reward: The external reward is the reward shared by all the agents given by the environment. Please refer to the Basic Setting section in Sec. 2: $r_{e}: S \times A_{1} \times A_{2} \times … \times A_{K} \to R$.

Perceived reward: Given this shared external reward, depending on a specific reward assignment, each agent can receive a perceived reward that drives its behavior. If the reward assignment is properly defined, then all the agents can act based on the perceived reward to jointly optimize (maximize) the shared external reward.

Local reward set: By defining $S_{local}$ to be the subset of states that affect heavily on each agent i’s optimal perceived reward, the agent $i$ can predict the optimal value function by only observing $S_{local}$. We then call this region the “local reward set” of agent $i$ (see Lemma 3 in the Appendix).

Definition of “nearby” agent: We define the distance between states $s_{1}$ and $s_{2}$ by the minimum number of steps the agent needs to transit from $s_{1}$ to $s_{2}$ under any possible policies. Following this definition, we similarly say that if the distance between agent 1’s state and agent 2’s state is small, they are nearby.

**We will make things more clear in the next revision.**

**References:**

[1] Wang, Weixun, et al. "From Few to More: Large-Scale Dynamic Multiagent Curriculum Learning." AAAI. 2020.

[2] Wang, Weixun, et al. "Action Semantics Network: Considering the Effects of Actions in Multiagent Systems." arXiv preprint arXiv:1907.11461 (2019).

[3] Jaques, Natasha, et al. "Social influence as intrinsic motivation for multi-agent deep reinforcement learning." International Conference on Machine Learning. PMLR, 2019.

[4] Devlin, Sam Michael. Potential-based reward shaping for knowledge-based, multi-agent reinforcement learning. Diss. University of York, 2013.

[5] Leibo, Joel Z., et al. "Multi-agent reinforcement learning in sequential social dilemmas." arXiv preprint arXiv:1702.03037 (2017).

---

### Decision · Program_Chairs · 2021-01-07
**Final Decision**

**Decision:**

Reject

**Comment:**

This paper proposes a method for collaborative multi-agent learning and ad-hoc teamwork. The paper includes extensive empirical results across multiple environments (including one of known outstanding high difficulty) and repeatedly performs favourably in comparison to a suitable set of state of the art methods. The proposed method is motivated by theoretical analysis, which was considered interesting but its connection to the method in the initial paper was weak.

Overall, there are remaining concerns which have not been fully addressed in the discussion phase. The authors' responses and discussion with the reviewers should be utilised to improve the material's presentation and to clarify the theory-empirical connection in future revisions of the paper.